# Exploring Consumer Preference towards the On-Farm Slaughtering of Beef in Germany: A Discrete Choice Experiment

**DOI:** 10.3390/foods12183473

**Published:** 2023-09-18

**Authors:** Josephine Lauterbach, Antonia Johanna Bruns, Anna Maria Häring

**Affiliations:** Faculty of Landscape Management and Nature Conservation, Eberswalde University for Sustainable Development, Schicklerstr. 5, 16225 Eberswalde, Germany

**Keywords:** consumer preferences, animal welfare, on-farm slaughter, slaughter conditions, premium beef marketing

## Abstract

Current production standards and communication campaigns about animal welfare in relation to beef strongly emphasise the “humane” rearing of cattle. Aspects such as transport and slaughtering conditions are often overlooked in both production standards and communications with consumers. Long transport routes and conventional slaughtering can cause significant stress to animals and have negative impacts on their welfare and on meat quality. On-farm slaughter can address these criticisms. Communicating the value of low-stress slaughtering conditions like on-farm slaughtering may offer significant sales potential for a premium market segment. In this study, we explore consumers’ preferences and willingness to pay for beef that is slaughtered on-farm rather than in conventional abattoirs. We conducted an online survey (*n* = 400) in 2022, with a sample that is representative of the German population with respect to gender, age, income and education. Our survey included a discrete choice experiment for the purchase of minced beef, incorporating product attributes that influence purchase decisions. These included: price, information on the social, economic and environmental benefits of regional production, different production standards (conventional/organic) and information on on-farm slaughtering. Our findings indicate that consumers derive the highest utility from a low price, followed by information about on-farm slaughtering. Participants indicated a preference for information on high animal welfare over high beef quality. We conclude that highlighting on-farm slaughtering could be a significant benefit in marketing premium beef products.

## 1. Introduction

In Germany, the issue of meat consumption and animal welfare has gained increased attention in both society and on the political agenda [1,2,3]. Numerous studies indicate that animal welfare is a central issue when purchasing meat products [4,5,6]. One often neglected, but key issue in animal welfare is animal transport and stressful slaughtering conditions [4,7,8,9]. Over the past decade, the meat processing and slaughtering industry in Germany has undergone radical change. Three companies now dominate the sector [10]. This consolidation has resulted in larger slaughtering facilities and animals having to travel further to be slaughtered [11].

Since 2021, revised EU regulations have facilitated on-farm slaughtering, which avoids animal transport and stressful slaughter at a conventional slaughterhouse [9]. These new regulations include fewer requirements for mobile slaughter units and an increase in the maximum transport time of the slaughtered animal to a stationary abattoir. This enables farmers to increase their production of “on-farm” slaughtered meat and to reach out to more consumers [12,13].

In this paper, we aim to gain more insights into consumer preferences concerning on-farm slaughtering and determine the willingness to pay (WTP) for on-farm slaughtered meat, using minced beef as a representative product. The issue was already addressed in a study in Sweden in 2007 [14] but should be kept under review in light of the changing meat market and new EU regulations.

We conducted a discrete choice experiment that presented prospective consumers with a variety of hypothetical purchase scenarios. We provided participants with key information on several quality attributes that might be critical in their purchase decisions. These quality attributes are summarised in the following section.

### 1.1. Quality Attributes in Meat Purchase Decisions

When assessing meat quality, one can differentiate four dimensions: product-oriented qualities, process-oriented qualities, quality control and user-oriented quality [15]. Other approaches differentiate between intrinsic and extrinsic quality cues, defining extrinsic quality as environmental, social and economic quality, while intrinsic quality includes sensory and nutritional qualities [15].

#### 1.1.1. Product Quality

Product quality or intrinsic quality encompasses various aspects such as sensory quality (e.g., taste and appearance) and health-related attributes (e.g., nutritional and sanitary quality) [15]. While consumers can usually assess these attributes, they may not always be able to do so during the purchase itself. Taste is generally considered the most crucial factor in consumers’ decision-making processes [16]. Other key sensory attributes include freshness and appearance [17,18]. Despite the negative health impacts associated with excessive consumption of red meat, beef is generally perceived as a healthy product [19]. Sanitary quality, such as the sales conditions (e.g., fresh vs. frozen) or the use-by date, can also be important purchase attributes [17].

#### 1.1.2. Process Quality

Process quality or extrinsic quality cues describe the way a product was produced, including animal welfare standards, a regional origin or the environmental impact of production [20]. Those attributes cannot be assessed by the consumer directly and are hence classified as “credence goods” [21].

High animal welfare standards can be considered to be the most important process quality aspect for consumers when buying beef and other meat products [17]. Consumers associate animal welfare with good animal husbandry, high animal wellbeing and appropriate feeding [4,22]. Animal welfare affects the perception of product quality, as consumers assume that high animal welfare positively influences the taste and nutritional value of meat [6]. Transport and humane slaughter have been considered to be less important to consumers when they think about animal welfare [4,23]. Linked to the issue of slaughter is the transportation of the animals to the abattoir, which many consumers have a negative view about [23].

Regional production enhances a wide range of quality aspects. These include economic quality, such as higher farm incomes through short supply chains and better payments to farmers and other value chain actors [24]. Regional production is also associated with higher product quality, as consumers expect better taste or fresher products [25,26]. Regional origin and its associated attributes may be considered less important in purchase decisions than product quality and animal welfare [17].

The environmental quality and impact of meat production include several factors such as greenhouse gas emissions, land use and impacts on biodiversity [15]. The environmental aspects of beef production have been found to be less important to consumers than product quality, animal welfare and regional production [17,27].

#### 1.1.3. Quality Control

The third quality sphere is quality control, i.e., a certain standard or classification of a product [20]. An important consumer-oriented production standard in Germany is the organic production standard, with a market share of about 16% of meat products [28]. The organic standard indicates high product and process quality. Animal welfare, environmentally sound production, fair working conditions, preservation of the landscape and biodiversity, as well as good taste and fewer additives all play an important role when buying organic food [5,29].

#### 1.1.4. User-Oriented Quality

User-oriented quality is subjective to the individual consumer and may include brand, purchase situation or price [20]. There are overlaps with other quality domains, as a local brand may trigger positive associations with regional production (process quality). The conditions of sale (i.e., the date of use, fresh vs. frozen) can be considered as both user-oriented and product quality.

Numerous studies have tried to determine the importance of price when buying meat. In general, unaided surveys and real-life experiments tend to show a high importance of price in the purchase decision [17,30]. For more conscious decisions, process quality such as animal welfare or regional origin is more important and can increase the willingness to pay for a meat product [27,31,32].

### 1.2. Transport and Slaughter

According to Council Regulation (EC) No 1099/2009, slaughter refers to the act of killing animals intended for human consumption. There are various methods available for slaughtering cattle. In the following sections, we describe both the conventional method of slaughtering used in most German abattoirs at present [33], as well as alternative slaughter methods.

Conventional slaughter consists of four stages: transport, pre-stunning, stunning and bleeding [8]. Transport refers to the entire process from the loading of the animals at the farm of origin to unloading at the destination. Typically, animals are transported by lorry or trailer to the slaughterhouse. The journey can take several hours, depending on the location of the slaughterhouse [34]. Economic pressure reduces the number of financially viable abattoirs, thus increasing travelling time [10]. Pre-stunning involves the arrival of the cattle and their transport to the stunning area. Stunning is the practice of rendering an animal unconscious or insensible to pain prior to slaughter. This is usually achieved by the use of stunning equipment such as captive bolt guns or electrical stunning [8,35]. After stunning, bleeding (or exsanguination) must occur within 60 s to ensure that the animal is killed without regaining consciousness. In commercial facilities, cattle are usually subjected to a process known as “chest sticking”, in which the arteries in the chest that supply oxygenated blood to the brain are cut, causing the animal to bleed to death [8].

Since 2021, revised EU regulations have facilitated alternative slaughtering [36]. In Germany, the following alternative slaughtering methods exist:Pasture slaughter: Cattle are stunned and killed on the pasture. Subsequent transport and processing take place in a stationary part of a slaughterhouse.Semi-mobile slaughter: Cattle are stunned with a captive bolt gun and then killed and bled on the farm of origin. Subsequent transport and processing take place in a stationary part of an abattoir.Fully mobile slaughter: Cattle are stunned with a captive bolt gun on the farm of origin. Slaughter and bleeding are carried out on site. All further processing takes place in an EU-approved mobile slaughter unit [37].

For pasture slaughter and semi-mobile slaughter, a maximum of three animals can be slaughtered per procedure. The carcass must be delivered to the stationary part of the slaughterhouse within 2 h.

#### Effects of Transport and Slaughter on Meat Quality

Transport and slaughter can affect several quality parameters, particularly product and process quality, which are interrelated. The presence of stress hormones can alter the taste and appearance of the meat. A particular problem is DFD meat (dark, firm and dry) which is difficult to commercialise. Long transport and waiting times at the slaughterhouse can increase the occurrence of DFD meat [38]. Therefore, stress and anxiety can affect product quality, while animal welfare is a key factor in process quality. Animals slaughtered on-farm or in local slaughterhouses show lower stress levels than conventionally slaughtered animals [9,39,40].

The use of mobile and on-farm slaughter methods can avoid the long-distance transport of live animals, thereby increasing animal welfare and contributing to improvements in economic returns (e.g., increased farm income), social quality (e.g., fair payment) and image quality (e.g., improved reputation for slaughter personnel). Nevertheless, mobile slaughter must also be viewed critically, particularly from the perspective of food safety. Appropriate measures (e.g., the presence of a veterinarian during the slaughter process, and short transport times to the stationary slaughterhouse for semi-mobile slaughter) have been included in the relevant EU regulation and its implementation in the individual EU member states [36,37].

## 2. Materials and Methods

### 2.1. Discrete Choice Experiment

The aim of discrete choice experiments is to create scenarios that mimic real-life market situations, where consumers decide whether or not to (hypothetically) purchase a product, based on the perceived utilities it offers [41,42,43,44]. Utility refers to the satisfaction or usefulness that an individual derives from consuming a particular good or service. The utility of a product is derived from the sum of all the component utilities of the product’s attributes, i.e., price, packaging, production standard, etc. [45]. McFadden’s Random Utility Theory states that consumers seek to maximise their utility when buying a certain product [46]. Discrete choice experiments can help researchers analyse the utilities of each attribute and determine the potential willingness to pay for each. This methodology has been successfully employed in previous studies on beef consumption [31,47,48,49].

Based on the literature overview presented in chapter 1, we identified four attributes each with up to four attribute levels to present participants with information that is relevant to their purchase decision (see Table 1).

In the study, participants were presented with 12 sets of choices, each containing three options of minced beef. We chose minced beef as it is a popular product in Germany, with which consumers are familiar [50]. To create a realistic market situation, each choice set offered two pre-defined options: the baseline option, which was the product commonly available at German supermarkets at the time of the study with no additional information and a price of EUR 4.99, and a no-buy option. The other two options in each set varied based on the attribute levels presented in Table 1. An example of a choice set is presented in Figure 1.

Price: We include price as a significant user-oriented quality. Moreover, price can be used to estimate the willingness to pay (WTP). The prices ranged from EUR 4.99/500 g to EUR 10.99/500 g. These prices were determined based on research conducted in supermarkets and organic shops in Northeastern Germany in July 2022. Some of the attributes of the minced beef offered in the choice experiment are not commonly available in German supermarkets (e.g., information on slaughter), so we set the highest price level higher than the observed market prices at the time. The price of EUR 4.99/500 g served as a baseline alternative.

Information on slaughter incorporates process quality and product quality. It describes the location where the slaughter occurred. If no information was provided, the (hypothetical) slaughtering happened in a conventional slaughterhouse, as is the most common practice in Germany. This level was taken as the baseline option for the attribute. Additional specific information was only provided when the livestock was slaughtered on-farm. To determine whether respondents were motivated by process quality or product quality, the attribute level of “slaughtered on-farm” was presented in two ways: one with the additional information of “no live transport” and the other with the additional information of “higher meat quality”.

Production standard was included as a user-oriented quality attribute. The attribute levels included no information, which suggested conventional production as the baseline option, and organic production as a second attribute level.

Information on the benefits of regional origin was included as another important process quality. The base level for this attribute was no information provided about the benefits of regional origin. The other levels of this attribute were determined by prior research [27,51], which identified “fair payment”, “transparent supply chain”, and “regional feed” as relevant qualities of regional production.

To ensure that the study reflected real-life purchasing conditions, the organic attribute was not paired with prices lower than EUR 8.99/500 g. Other attribute combinations were omitted for similar reasons. The highest price option was only offered in combination with the organic label and information about both the location of slaughter and the benefits of regional origin. In total, there were 23 possible attribute level combinations, in addition to the baseline product (Appendix B). The 12 choice sets were created randomly using a random number generator and manually assembled in an Excel sheet before being imported to QuestionPro.

### 2.2. Data Analysis

Hensher, Rose, and Greene [43] proposed that the utility (*U*) a person (*n*) derives from selecting an option (*j*) in a given scenario (*s*) and is made up of both observable utility (*V*) and an unobservable element (*ε*):(1)$$U_{nsj}=V_{nsj}+\varepsilon_{nsj}$$

It is assumed that the unobservable component (*ε_nsj_*) is randomly distributed. The observed utility (*V_nsj_*) is the sum of all the linear combinations of the observed attribute levels (*x*) and their respective parameters (β):(2)$$V_{nsj}=x{ß}_{slaughtering\;no\;live\;transport}+x{ß}_{slaughtering\;meat\;quality}+x{ß}_{r.origin\;fair\;payment}+x{ß}_{r.origin\;fair\;payment}+x{ß}_{r.origin\;feed}+x{ß}_{price}$$

We analysed the choice experiment by running a Cox regression in IBM SPSS 29. Cox regression is based on a multinomial logit choice model which provides the relative importance of each utility attribute. Cox regression can be used to analyse discrete choice experiments by modelling the choice of an individual as a time-to-event outcome. To analyse the data with a Cox regression, the alternatives in each choice set are considered as different treatments and the time-to-event outcome is the time at which individuals make their choices. The attributes of the alternatives are then included as predictor variables in the Cox regression model. All the attribute levels were treated as dummy variables with the baseline serving as the reference category. The coefficient estimates for these variables represent the effect of each attribute on the choice, controlling for the effects of the other attributes [41]. Cox regression modelling is commonly used in medicine [52] but has also been applied to determine consumer preferences [53,54]. 

Using the utility parameters (β) we determined the relative importance of each attribute (*l*) in the hypothetical purchase situation to evaluate the extent to which each attribute contributes to the overall utility of the product. The attribute importance can be calculated by determining the utility range of a specific attribute (*j*) and then dividing it by the utility range of all attributes presented in the study: $\overset{J}{\underset{j}{\sum }}\left({ß}_{j1}-{ß}_{j2}\right)$. Lastly, the results were normalised [55]:(3)$$l=\frac{\left({ß}_{j1}-{ß}_{j2}\right)}{\sum_{j}^{J}\left({ß}_{j1}-{ß}_{j2}\right)}*100$$

The results of the modelling were used to estimate the willingness to pay (*WTP*) for specific attribute levels. We determined the *WTP* for each attribute using the mean value of all the attribute levels of a specific attribute. For this purpose, the following formula was applied [56]:(4)$$WTP=-\frac{{ß}_{j}}{\left({ß}_{price1}-{ß}_{price2}\right)}\;$$

### 2.3. Sampling

We collected data using an online survey (*n* = 400), representative of the German population according to gender, age, income and education (based on [57]). Participants were recruited via an online panel provider (Bilendi respondi) and received an incentive for participating. Only people who consume beef at least once a month were eligible for the survey. The questionnaire consisted of five sections: an introduction, a data protection agreement, screening questions concerning beef consumption, socio-economic and demographic questions, a discrete choice experiment and questions regarding knowledge of slaughter. The questionnaire was pre-tested with 50 participants after which minor revisions were made.

## 3. Results

Of the 400 respondents who completed the questionnaire, 396 respondents were included in the analysis. The remaining participants were excluded due to data quality issues or technical errors [58,59]. Table 2 summarises the key demographics of the sample. The sample can be considered representative of the German population in terms of age, gender, educational qualifications and net household income [57]. Participants from all 16 federal states took part in the study: of which 15% live in the former East and 85% in the former Western federal states, which constitutes a slight overrepresentation of the latter.

The majority of the participants stated that they ate beef once or twice a month (37%), while around 31% ate beef more often. The most common place for people to purchase beef is at the supermarket, with 35% buying fresh products and 33% of respondents choosing from self-service counters. A smaller proportion of participants (23%) buy beef at a butcher’s shop, while a minority prefer to purchase beef at organic supermarkets (4%), farmer’s markets (3%), or elsewhere (1.5%). The majority of respondents (70%) reported that they make purchasing decisions not only for themselves but also for other household members, while 30% only buy food and other items for themselves.

Around 2/3 of the participants stated that they had not previously dealt with the topic of slaughter, either actively or passively. For those who have dealt with the topic before (31%), different factors played a role, e.g., the way they informed themselves or the circumstances under which they acquired their knowledge. Active engagement with the topic of slaughter includes deliberate information seeking, such as watching documentaries, visiting butcher shops, or encountering the topic in a professional setting. Those participants stated their interest in animal welfare and that they wanted to know how animals are dealt with during transport and slaughter. Passive engagement, on the other hand, refers to cases where participants have not intentionally sought information but have nevertheless come into contact with the topic, for example, through childhood memories or other chance encounters. A total of 51% of the participants stated that they are interested in learning more about slaughtering, while 45% are not.

The choice experiment was analysed using Cox regression, which was used to estimate part-worth utilities and odd rations for the different attribute levels with a chi-square value of 1,918,810 and degrees of freedom = 10 (*p* < 0.01). The results of the Cox regression are presented in Figure 2. The values were highly significant for the attributes of price, information on slaughter and production standards (*p* < 0.01) while for information on regional benefits, only the attribute levels of fair payment and a transparent supply chain were significant (*p* < 0.05). The attribute level of the regional feed was not significant (*p* = 0.429). All values were interpreted in relation to the baseline product (EUR 4.99/500 g, no information) which was set at 0. Additionally, we could estimate the attribute importance according to Formula (3).

Figure 2 shows the importance of different attributes in the hypothetical purchase situation the participants faced. Price is clearly the most important attribute, for 57% of respondents. The utility of this attribute decreases as the price increases, so the baseline price has the highest utility and the highest price of EUR 10.99 has the lowest utility. Information on slaughter was the second most important attribute. Here, the process quality-oriented option (no live transport) had a higher utility value than the product-oriented option. Overall, the utilities for on-farm slaughter were the highest. Production standards were less important and their part-worth utility scores were lower than for on-farm slaughter. Information about the benefits of regional production was perceived as least important in the purchasing decision, with the utility scores for these being very low. The no-buy option was only chosen in 1% of all choice situations and has a negative part-worth utility.

The odds ratio is a measure of the likelihood of a hypothetical purchase occurring (dependent variable) given certain attribute levels (independent variables) compared to the odds of the dependent variable occurring in the absence of the independent variable [41]. A value of 1 indicates no association between the two variables, while a value greater than 1 indicates a positive association (i.e., the product attribute increases the odds of hypothetically buying the product) and vice versa for a value less than 1. The odds ratios presented in this study show similar trends as the part-worth utility scores, i.e., the likelihood of buying the product decreases with increasing prices. The results also indicate that information on on-farm slaughter makes it more than twice as likely that participants choose the product. Organic production also increases the likelihood by a factor of 2, while information on regional benefits does not influence the hypothetical purchase situation.

Table 3 shows the marginal WTP for all attribute levels according to Formula (4). It can be understood as an indicator of the relative importance of the attribute levels shown in this choice experiment, but not as actual market prices, as not all relevant information was presented to the participants.

The WTP reveal similar trends as the utility scores and odds ratios. The findings indicate that consumers are generally willing to pay a premium price of EUR 2.26 for the provision of “information on slaughter” with the specification of “no live transport”, and a slightly lower premium price of EUR 1.96 for “higher meat quality”. Consequently, communicating on-farm slaughtering results in an average WTP premium of EUR 2.11. Consumers expressed a WTP at a premium price of EUR 1.63 for organically produced products. The WTP for “information about the benefits of regional origin” was relatively lower, averaging EUR 0.51, and ranging from EUR 0.71 for “fair payment” to EUR 0.23 for “regional feed”.

## 4. Discussion

The objective of this study was to gain more insights into consumer preferences concerning on-farm slaughtering and determine willingness to pay (WTP) for on-farm slaughtered meat, using minced beef as a representative product.

As shown in other studies, price was the most influential factor in the hypothetical purchase decision survey participants faced. A higher price decreases the utility of the product [14,17].

The second most important attribute was information on slaughter. This finding contradicts other studies suggesting that consumers do not want to actively deal with the topic of slaughter [17,60,61]. This can be explained by the cognitive dissonance of caring about animals on the one hand, but harming them through slaughter and consumption on the other hand (the “meat paradox”) [62].

To avoid cognitive dissonance consumers can adopt three strategies: (1) changing their values, i.e., valuing animal welfare less, (2) changing behaviour, i.e., eating less meat or more animal-friendly meat or (3) obscuring the contradiction between one’s values and behaviour, i.e., avoiding thinking about slaughter [62,63]. Buying meat from on-farm slaughter can be understood as applying strategy 2 (changing behaviour) and strategy 3 as a way of reducing the conflict between values and behaviour. People’s reluctance to engage with the topic of slaughtering might explain why the attribute level “no live transport” indicating higher animal welfare standards was slightly more important than better meat quality. Moreover, high animal welfare might already be associated with higher product quality [64] making the specification “higher meat quality” redundant.

The high utility for on-farm slaughter could also be based on a need for knowledge on the subject of slaughtering, as around 50% of participants stated in this study. Currently, information on slaughtering is not generally given in consumer communications. As a result, consumers are generally unable to base their purchasing decisions on the slaughtering process. This creates a considerable knowledge deficit, and even informed consumer groups, such as organic consumers, know little about slaughter [65]. This deficit should be addressed by value chain actors, such as producers or the processing industry by providing more information on slaughtering.

Neutral information can increase consumer trust and is considered an important aspect of consumer choice [21,66,67]. As the slaughtering industry has a bad reputation in Germany [1], conveying information on on-farm slaughtering might improve the overall perceptions of the quality of the product. A high willingness to pay for the mobile slaughter of cattle has already been demonstrated in another hypothetical purchase experiment [14]. However, a higher WTP cannot be generalised for all animal species (e.g., the WTP for chicken is lower than for minced beef [14]) and such theoretical considerations of consumers do not always translate into actual changes in behaviour at the point of sale [30,44,68].

The utility and odds ratio for organic production was lower than for on-farm slaughtering. This is quite unexpected because numerous consumers link organic production with additional qualities included in the choice experiment, such as favourable working conditions, fair wages, and even on-farm slaughtering [5,65]. However, the EU regulation for organic farming does not specify (on-farm) slaughter [69]. The absence of specific regulations for the slaughter of organic animals carries the risk of organic consumers feeling deceived. This could cause them to question the rationale behind consuming organic meat, as it serves as a justification to consume meat to overcome the meat paradox. Hence, it might be especially important for the organic sector to communicate neutrally on the issue of slaughtering and for policymakers to include on-farm slaughtering in organic regulations [65].

The benefits of regional production achieved the lowest utility values and associated odds ratio and willingness to pay. This may be due to consumers considering other attributes such as organic production or on-farm slaughtering as already meeting their needs [5]. The attribute level “fair payment” received the highest utility within this attribute. Fair payment can be interpreted as a process quality as it suggests that producers or other actors in the supply chain benefit from consumers’ purchasing decisions by receiving higher incomes. This attribute may have been influenced in particular by media reports on the poor working conditions in conventional slaughterhouses during the COVID-19 pandemic [48,70]. In this case, it would also have been interesting to consider “fair payment” as an attribute level of on-farm slaughtering and evaluate whether its utility would increase. Labelling a product with information on a transparent supply chain can provide at least some utility by addressing consumers’ demand for product quality-related issues, such as food safety and health [25]. Such transparency enables consumers to trace the origin of their food, thereby increasing their trust in food quality and safety [26,67]. In the meat sector, food traceability is especially difficult [71], and scandals surrounding the safety and production of meat products in recent decades have frequently undermined consumer trust in such products [1].

Information about “regional feed” provides the lowest utility within this attribute. This contrasts with previous studies that showed a strong preference for local feed, at least within the organic sector [51,72].

This study is not without its limitations. Due to technical difficulties, the 12 choice sets were presented in the exact same order for each participant and could not be randomised. This may have resulted in sequence effects that favour certain positions in the experiment design. Respondent fatigue could have contributed as well [41]. Additionally, the sample size of 400 is rather small.

Finally, when interpreting the results, one should consider the hypothetical background of this experiment [44]. We can assume that social desirability effects exist [73,74] and not all aspects that influence the purchase decision were tested, due to methodological considerations (e.g., use by date, fresh vs. frozen [4,17]). In addition, we cannot assume that all the attributes presented were actually noted by the participants. A combination with an eye-tracking study would be useful here to assess which information is actually perceived by consumers [75]. Furthermore, some attribute levels might have been redundant for the participants based on their own associations, e.g., on organic farming. An in-depth qualitative study focusing solely on the associations and perceived benefits of on-farm slaughtering (also for other meat than beef) could usefully be the focus of further research.

## 5. Conclusions

In conclusion, it is essential that value chain actors, including producers and the processing industry, take proactive measures to address the significant lack of consumer knowledge about the slaughter process by providing comprehensive information about slaughter. This would not only improve consumer understanding but also potentially influence their purchasing decisions and overall perception. In spite of the limitations of this study [73,74] and the potential sensitivity surrounding the topic of animal slaughter in consumer discourses, our study shows a robust interest and willingness to pay for meat slaughtered on the farm. Nevertheless, it is important to note the significance of price, and the consequent reduction in utility, as prices increase. In terms of marketing strategy, meat slaughtered on-farm should initially be targeted at consumers who are less price-sensitive and are in the premium segment of the market. Communication of the benefits should primarily emphasise increased animal welfare.

## Figures and Tables

**Figure 1 foods-12-03473-f001:**
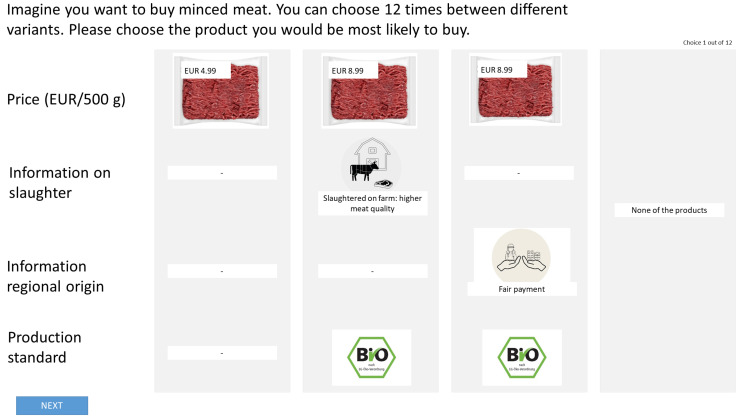
Example of a choice set (Own illustration according to Questionpro output). The baseline and no-buy options were placed first and last while the other two options varied for each choice.

**Figure 2 foods-12-03473-f002:**
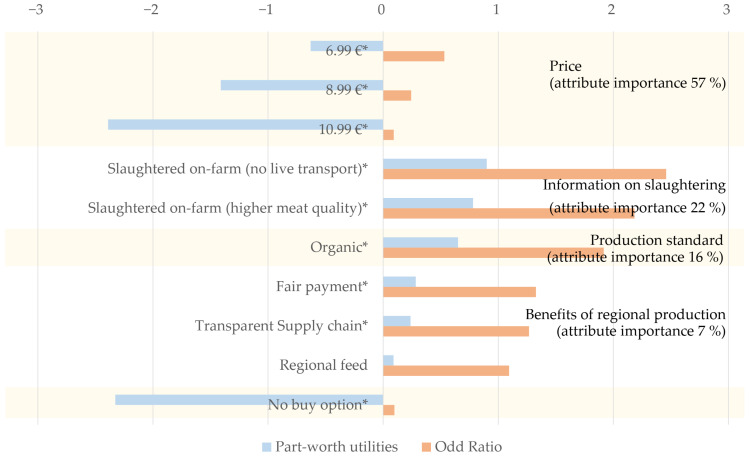
Results of the conjoint analysis using Cox regression, baseline scenario (EUR 4.99, no further information) set to 0; * *p* < 0.05.

**Table 1 foods-12-03473-t001:** Overview of different attributes and attribute levels used in the choice experiment.

Attribute	Quality	Level			
		1	2	3	4
Price (EUR/500 g)	User-oriented quality	4.99	6.99	9.99	10.99
Information on slaughter	Process quality, product quality	No information	Slaughtered on farm, no live transport	Slaughtered on farm, higher meat quality	--
Production standard	Quality control	No information	Organic production	--	--
Information on benefits of regional origin	Process quality	No information	Fair payment	Transparent supply chain	Regional feed

**Table 2 foods-12-03473-t002:** Key demographics ^1^.

Key Demographic	Distribution in %
Gender	Male: 50.3; Female: 49.3; Diverse: 0.3
Age in years	18–24: 10.1 25–34: 14.6 35–44: 15.2 45–54: 18.7 55–64: 17.2 35–64: 51 65 and over: 24
Education	No vocational qualification: 3.51 Currently in school or vocational training: 10.78 Completed vocational education: 51.63 Academic degree: 33.08
Monthly net household income in EUR	Under 1000: 7.8 1000–2000: 25.5 2000–3000: 24 3000–4000: 16.4 4000–5000: 10.6 Over 5000: 13.4 1500–3500: 46.25 Over 3500: 31.2
Size of municipality	Under 5000: 16.29 5000–10,000: 18.05 10,000–100,000: 22.81 100,000–500,000: 15.79 Over 500,000: 21.8

^1^ Participants answering “no information” are not included.

**Table 3 foods-12-03473-t003:** Willingness to pay (WTP) for different attribute levels.

Attribute	Attribute Levels	WTP for Attribute Levels	Average WTP for Attribute
Information on slaughter	Slaughtered on-farm (no live transport) Slaughtered on-farm (higher meat quality)	EUR 2.26 EUR 1.96	EUR 2.11
Production standard	Organic	EUR 1.63	EUR 1.63
Information on benefits of regional origin	Fair payment Transparent supply chain Regional feed	EUR 0.71 EUR 0.60 EUR 0.23	EUR 0.51

## Data Availability

Data are contained within the article or Appendix A.

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
