# Peer review of "Exploring Consumer Preference towards the On-Farm Slaughtering of Beef in Germany: A Discrete Choice Experiment"

_foods, 2023, doi:10.3390/foods12183473_

Round 1
Reviewer 1 Report
Review ref. Foods 2588102
The work entitled Exploring Consumer Attitudes towards the On-Farm Slaughtering of beef in Germany: A Discrete Choice Experiment? (ref. Foods-2588102) has been submitted for review.
The document is important to the foods science. There is a great experimental effort in the document and in the project of the research team.
Currently the consumer is motivated by the welfare of animals and the consumption of meat responsibly. Governments are also aware of this idea and must gather the concerns of the population. For this reason, the document presented could contribute to the development of the meat production sector. Socially it is very important. Methodologically, the document is correct, although it suffers from some issues related to the approach to animal welfare and the health implications of slaughtering animals in a slaughterhouse.
Correct experimental design. However, there is a lack of direct questions to the consumer about the advisability of slaughtering cattle at the place of production and the implications for the health of animals and humans that this entails. I consider that there are no references about the opinion of the consumer about the welfare of the animals and about the guarantees of the traceability of the consumption of this meat. Is the consumer aware of the health organization implications of slaughtering animals in a slaughterhouse?
Could the authors focus the document on a panel of animal welfare experts? Then the consumer survey could be done.
The survey is well structured, but questions about the welfare of the animals at the time of slaughter are missing. This is a point of view that concerns the consumer.
The document requires citing previous experiences in terms of animal welfare at slaughter related to the transport of animals to the slaughterhouse.
The document must be rejected in its current format. Too many uncertainties are detailed in the paper that the authors are aware of. The document can be resubmitted. The authors should make an effort to focus on consumer opinion about animal welfare. The introduction of must be worded differently and a focus on animal welfare must be taken up and the health implications for humans.
I am aware that the survey has been carried out and that this may be a difficulty when writing a new document.
Some suggestion is based on the following:
Title: It is correct.
Abstract:
Line 10: reference to ‘outdoor areas for cattle’ is not in relationship with the aim of this document. That is important in an abstract. In discussion section it could be included.
Line 14: ‘In Germany, the slaughtering sector is highly consolidated’. What is the meaning of this sentence? What is ‘consolidated’? Maybe you are having reference to solid regulations. It is good, then you should included a reference.
Lines 20-21: The aim of the document could be included.
Introduction
The introduction should be rewritten. Before the aim of study the importance of animal welfare for the consumer must be mentioned.
Line 41: feeding)
Lines 43 and 44: What about the changes? What about sector? Maybe “meat “? Must be more specific.
Sections 1.1 and 1.2 should be Material and methods section.
Lines 68-80 are not neccessary. Probably they could be used in discussion of results. That is not a review document. You must You should synthesize the meaning of “Product quality”, “Process quality”, “Quality control” and “User-oriented quality”.
Lines 155-167 should be delected.
Lines 174-186 are important for the introduction because they show the background of the document
Lines 187-198: they should be summarized
Materials and Methods
Line 201: “……it offers [54–56], [57] (p 6)”. We are not able understand it.
Lines 199-209: they should be summarized
Lines 223-224: “If no information was provided, it was assumed that the slaughter happened in a conventional slaughterhouse”. That is confused. Why is it not "slaughtered in a conventional slaughterhouse"? For example.
Figure 1. Example of a choice set (in German). You should be more specific in the figure caption
The presentation of the sets to the consumer is confusing.
I miss a classic question in this type of survey: ‘Would you be willing to pay more for this product compared to the industrial slaughter of animals?’
Would you be willing to pay more for this product compared to the industrial slaughter of animals?
What kind of response is expected from the consumer? Quantitative, qualitative answer?
Results
Line 347: This sentence should be clarified.
Table 4: The abbreviation WTP should be clarified in the table
Line 369-377: These are a discussion.
Discussion
Line 383: What about utility?
Lines 384-388: However, "the cognitive dissonance of caring about animals on the one hand, but harming them through slaughter and consumption on the other hand" is the second most important attribute. Could be explain?
Lines 389-397: Could you clarify if the consumer values the organoleptic quality of the meat or the welfare of the animals? It is not clear from the wording.
Lines 399-404: I do not understand the relationship between the results of the document and the author references listed in lines 399 to 404.
Line 409: “A high willingness to pay for mobile slaughter of cattle has already been demonstrated in another hypothetic purchase experiment” Then… That is not a novel document? References about that document could be reported in order to expand knowledge. We can’t forgot this is the aim in the present work.
Lines 423 and 425: “Hence, it might be especially important for the organic sector to communicate neutrally on the issue of slaughtering and for policy makers to include on- farm slaughtering in organic regulations. This sentence is not a discussion. That could be a conclusion.
Line 434: The reference to Covid 19 is inconsistent in this document. Consumers have been expressing their disapproval of the slaughter of animals in the slaughterhouse regardless of the Covid 19 pandemic.
Lines 443-450: I believe that this issue has little to do with the reason for the work presented. The document is focused on the convenience of slaughtering animals on the farm to avoid stress on the animals fundamentally. In this sense I do not understand the reference to the type of food that the animals have received. This may affect consumer response. Extensive references in the literature have dealt with the objective of the convenience of using local foods in animal feed and local trade.
Lines 451-454: This is true and the authors correctly reflect on the difficulties of this type of study based on online surveys. This is very important. Reference about this problem could be included in the conclusion.
Line 461: “A combination with an eye-tracking study would be useful here”. Your reflection on the limitations of the study is correct. This probably requires another experimental approach with direct and simple questions to the consumer. A doubt arises. What would be the ocular comparison? Present meat samples to consumers? This is not the reason for work. It is not about assessing the organoleptic characteristics of meat slaughtered nearby. The document aims to assess the sensitivity of the consumer about the slaughter of animals in the place of production.
Lines 463-465: This sentence is better as a conclusion.
Conclusions
The conclusions could be accepted. However the results about the opinion of animal welfare are not supported by the conclusions.
References to the limitations of the study should be indicated in the conclusions.
References:
85 references is too many for the document.
Important references of the year 2022 are missing. See Guarnido-López et al. (2022). Livestock Science,259, 104904. https://doi.org/10.1016/j.livsci.2022.104904
Reviewer 2 Report
The research article with the title "Exploring Consumer Attitudes towards the On-Farm Slaughtering of beef in Germany: A Discrete Choice Experiment" has 19 pages, 85 references, 4 Tables, 1 Figure and 1 Appendix.
The article has a fairly simple hypothesis, which its authors confirmed.
I have serious comments that need to be perceived especially from a formal point of view.
The abstract is too extensive, it has 21 lines. However, it does not contain essential information about the experiment, such as confirmed (probability) results and a conclusion in terms of the importance of the topic.
Keywords are used from the title, and it would be better to choose those that support better searchability in the database, which is the purpose of keywords.
Introduction
A large number of references are used in the introduction of the work, but they are used for their own purposes. It is a total of 53 references, which are out of a total of 85. That is 62%. However, they are given for no particular reason, just to be there. the example on lines 38 to 41 is used in two sentences 8 references:
Example: This can be seen as a premium market, which has been explored by several studies [6–10]. Within this premium market segment an important aspect when buying meat is a high animal welfare standard (e.g. outdoor access, low stocking density, high quality feeding [11–14].
The content of this chapter is not bad, but it gives the impression of using some previous research/work.
Material and Methods
What the introduction chapter is rich in references, this chapter is, on the other hand, poorer.
The text in table 1 is confusing and should be better formatted.
However, the information is comprehensible and it would be better to link it with other comparable research through the missing references.
Results
This chapter is based on three tables where the results are presented, including their statistical evaluation. However, nowadays it is much better to use the graphical options for expressing results using graphs, which is missing here and reduces readability and potential attractiveness for readers.
Typo: Table 4: € 0.,51
Discussion
The discussion is fairly well done with the findings, but there is little that can be read as recommendations for the various parties involved in the issue. Understanding only the consumer and the seller is insufficient. There are many more chain members involved (state, producer/farmer...).
Conclusion
Conclusion is too general and again does not perceive other links of the chain.
References
Not sure if all references are necessary.
Reviewer 3 Report
The manuscript foods-2588102 entitled "Exploring Consumer Attitudes towards the On-Farm Slaughtering of beef in Germany: A Discrete Choice Experiment”
The manuscript is well-written, and informations and the contents of the manuscript are of great value. However, there are some areas that require further improvement to enhance the overall quality of the paper
1) I will suggest the authors to add one more sub-section in the introduction part regarding the impact of long transportation distance on meat quality attributes such as color, tenderness etc., with a particular focus on the important meat quality problems linked with pre-slaughter stress. DFD & PSE. The authors just touched this topic at line 187. But the addition of detailed discussion with reference to the pH changes, protein denaturation, opening of muscle structure etc.
2) Discuss the incidence of these two problems, particularly focusing Germany and also discuss it with reference to other regions of the world. How these conditions may affect purchasing decisions of consumers due to unpleasant color and purge or drip loss in meat packs.
3) Also, include discussion of on-farm successful models if any? Implemented in other countries. It will further add value to the manuscript.
4) Additionally, I suggest to add a table or figure indicating, advantages and disadvantages of the on-farm slaughtering, particularly, for food safety point of view and environmental pollution. This Table or Figure should have a follow up discussion on this topic. How we can ensure implementation of food safety standards on farm level slaughtering.
5) Figure 1. Please improve figure quality by increasing size and DPI.
The conclusion section could benefit from incorporating personal insights or providing a future perspective.
Round 2
Reviewer 1 Report
The authors have made the suggested improvements. Review the formal aspects of the magazine.
Reviewer 2 Report
Based on the modifications of authors, I have no other comments.
Reviewer 3 Report
I am satisfied with authors response.